# One-Step Soft Agar Enrichment and Isolation of Human Lung Bacteria Inhibiting the Germination of *Aspergillus fumigatus* Conidia

**DOI:** 10.3390/microorganisms12102025

**Published:** 2024-10-07

**Authors:** Fabio Palmieri, Jérémy Diserens, Manon Gresse, Margo Magnin, Julina Helle, Benoît Salamin, Lorenzo Bisanti, Eric Bernasconi, Julie Pernot, Apiha Shanmuganathan, Aurélien Trompette, Christophe von Garnier, Thomas Junier, Samuel Neuenschwander, Saskia Bindschedler, Marco Pagni, Angela Koutsokera, Niki Ubags, Pilar Junier

**Affiliations:** 1Laboratory of Microbiology, Institute of Biology, Faculty of Science, University of Neuchâtel, 2000 Neuchâtel, Switzerland; jeremy.diserens@unine.ch (J.D.); manon.gresse@student.uliege.be (M.G.); margo.magnin@unine.ch (M.M.); julinahelle10@gmail.com (J.H.); benoit.salamin@unine.ch (B.S.); lorenzo.bisanti@unine.ch (L.B.); saskia.bindschedler@unine.ch (S.B.); 2Division of Pulmonary Medicine, Department of Medicine, Lausanne University Hospital (CHUV), University of Lausanne (UNIL), 1066 Epalinges, Switzerland; eric.bernasconi@chuv.ch (E.B.); julie.pernot@chuv.ch (J.P.); apiha.shanmuganathan@chuv.ch (A.S.); aurelien.trompette@chuv.ch (A.T.); christophe.von-garnier@chuv.ch (C.v.G.); angela.koutsokera@chuv.ch (A.K.); 3Vital-IT, Swiss Institute of Bioinformatic (SIB), 1015 Lausanne, Switzerland; thomas.junier@sib.swiss (T.J.); samuel.neuenschwander@sib.swiss (S.N.); marco.pagni@sib.swiss (M.P.)

**Keywords:** lung microbiome, bronchoalveolar lavage fluid (BALF), antagonistic bacteria, biocontrol, *Pseudomonas aeruginosa*, aspergillosis

## Abstract

Fungi of the genus *Aspergillus* are widespread in the environment, where they produce large quantities of airborne conidia. Inhalation of *Aspergillus* spp. conidia in immunocompromised individuals can cause a wide spectrum of diseases, ranging from hypersensitivity responses to lethal invasive infections. Upon deposition in the lung epithelial surface, conidia encounter and interact with complex microbial communities that constitute the lung microbiota. The lung microbiota has been suggested to influence the establishment and growth of *Aspergillus* spp. in the human airways. However, the mechanisms underlying this interaction have not yet been sufficiently investigated. In this study, we aimed to enrich and isolate bacterial strains capable of inhibiting the germination and growth of *A. fumigatus* conidia from bronchoalveolar lavage fluid (BALF) samples of lung transplant recipients using a novel enrichment method. This method is based on a soft agar overlay plate assay in which bacteria are directly in contact with conidia, allowing inhibition to be readily observed during enrichment. We isolated a total of five clonal bacterial strains with identical genotypic fingerprints, as shown by random amplified polymorphic DNA PCR (RAPD–PCR). All strains were identified as *Pseudomonas aeruginosa* (strains b1–b5). The strains were able to inhibit the germination and growth of *Aspergillus fumigatus* in a soft agar confrontation assay, as well as in a germination multiplate assay. Moreover, when compared with ten *P. aeruginosa* strains isolated from expectoration through standard methods, no significant differences in inhibitory potential were observed. Additionally, we showed inhibition of *A. fumigatus* growth on Calu-3 cell culture monolayers. However, the isolated *P. aeruginosa* strains were shown to cause significant damage to the cell monolayers. Overall, although *P. aeruginosa* is a known opportunistic lung pathogen and antagonist of *A. fumigatus*, we validated this novel one-step enrichment approach for the isolation of bacterial strains antagonistic to *A. fumigatus* from BALF samples as a proof-of-concept. This opens up a new venue for the targeted enrichment of antagonistic bacterial strains against specific fungal pathogens.

## 1. Introduction

According to the most recent estimates, 3.8 million people die from fungal infections every year worldwide [1]. This is notably the case for fungi belonging to the genus *Aspergillus*, which are ubiquitous in the environment [2], and whose conidia can reach a density of up to 10^8^ per m^3^ of air [3]. *Aspergillus* spp. are opportunistic pathogens affecting 14 million people worldwide [4] and are associated with a wide spectrum of diseases, called aspergillosis. The most common causative agent of aspergillosis, *A. fumigatus*, has been recently classified as one of the four fungal pathogens of critical importance in the first fungal priority pathogens list published by the WHO to guide research and public actions [5]. The clinical manifestations of aspergillosis range from hypersensitive reactions to lethal invasive infections, depending on the immune status of the host [6,7]. While conidia are cleared either via mucociliary movement or phagocytosis by alveolar macrophages in an immunocompetent setting, this is not the case in an immunocompromised host, where conidia are not adequately cleared, thus leading to the outgrowth of *Aspergillus* in the lungs [6,7]. Once inside the lung, conidia are confronted with microenvironmental conditions, such as altered pH and iron bioavailability, or hypoxia, which can impact their germination and growth [8,9,10,11,12,13,14]. 

While aspergillosis development has been widely shown to be influenced by host and microenvironmental conditions, the specific contribution of the lung microbiota as a major factor influencing colonization and growth of *Aspergillus* spp. has not been sufficiently addressed. Hérivaux and colleagues [15] showed that a lung microbiota community structure characterized by an increased abundance of the bacterial genera *Staphylococcus*, *Escherichia*, *Paraclostridium*, and *Finegoldia* can be associated with the development of invasive pulmonary aspergillosis (IPA) in a cohort of 104 critically ill immunocompromised subjects. Recently, another study, led by Ao et al. [16], revealed that the lung microbiota, particularly *Streptococcus salivarius*, *Prevotella timonensis*, and *Human betaherpesvirus 5*, is strongly associated with laboratory biomarkers linked to inflammation and the immune response (i.e., total lymphocytes counts, serum albumin, or serum lactate dehydrogenase), and clinical outcomes in community-acquired pneumonia patients with IPA. However, the underlying mechanisms explaining the association between the lung microbiota and disease outcome in patients suffering from invasive aspergillosis still needs further investigation. The interaction with bacteria inhabiting the lungs might also act as a factor affecting conidia germination and subsequent fungal growth [7,17]. 

The airway microbiome is a highly complex ecosystem composed of diverse microorganisms, including bacteria, fungi, viruses, and archaea [18]. These microorganisms interact with each other and the host, influencing various physiological and pathological processes [19,20]. The composition and dynamics of the airway microbiota are shaped by numerous factors, including the host’s immune status, environmental exposures, as well as antimicrobial treatments [21]. These interactions can significantly impact the colonization and growth of pathogens such as *Aspergillus* spp. The interkingdom interactions between bacteria and fungi in the airway are particularly noteworthy, as they can modulate the pathogenicity of fungal species [22]. For instance, bacterial metabolites and biofilm formation can influence fungal growth and virulence, while fungal presence can alter bacterial community structure and function [22,23].

The inherent complexity of the airway microbiota poses significant challenges for the investigation of the interaction between *A. fumigatus* and antagonistic bacteria, and ultimately the isolation of such bacterial strains. Indeed, traditional isolation methods that involve culturing bacteria in isolation before testing them against fungal pathogens may fail to capture the intricate interactions occurring within the lung microbiome. These interactions often involve multiple species and can be influenced by physical and chemical factors of the airway environment [22,23]. Consequently, a more integrative approach that considers the bacterial–fungal interaction context is essential for the enrichment and isolation of bacterial strains with fungal inhibiting properties. 

The aim of this study was to enrich and isolate bacterial strains capable of inhibiting the germination and growth of *A. fumigatus* conidia from bronchoalveolar lavage fluid (BALF) samples of subjects who had undergone lung transplantation. We utilized a novel enrichment method based on a soft agar overlay plate assay, in which bacteria from BALF samples were put directly in contact with *Aspergillus* conidia. This approach allowed us to enrich and isolate five antagonistic bacterial strains that inhibited the germination of *A. fumigatus* conidia. All isolated strains were identified as *Pseudomonas aeruginosa*, with their inhibitory effect observed during the enrichment by the presence of an inhibition halo. Additionally, we compared the genotypic diversity of the five isolated strains, with ten P. aeruginosa strains isolated from expectoration through standard methods and their inhibitory effect using a high-throughput germination screening assay. Finally, two of the five isolated P. aeruginosa strains were selected for subsequent co-culture confrontation assays on Calu-3 cell culture monolayers.

## 2. Materials and Methods

All methodological steps outlined below (Section 2.1, Section 2.2, Section 2.3, Section 2.4, Section 2.5, Section 2.6, Section 2.7, Section 2.8, Section 2.9, Section 2.10, Section 2.11, Section 2.12, Section 2.13 and Section 2.14) are illustrated in the flow chart diagram depicted in Figure 1.

### 2.1. Culturing of Fungal and Bacterial Strains

The *A. fumigatus* strains used in this study are summarized in Table 1. The strains were routinely cultured on malt extract agar (MEA) plates composed of 30 g/L malt extract (Sios Homebrewing GmbH, Wald, Switzerland), 5 g/L casein peptone (LLG Lab Logistics Group GmbH, Meckenheim, Germany), and 15 g/L technical agar (Biolife Italiana, Milano, Italy). Plates were incubated at room temperature for 10 days. The wild-type (WT) isolate was obtained from the Westerdijk Fungal Biodiversity Institute—KNAW (CBS 144.89). This strain is an isolate with a clinical origin and is used extensively due to its higher pathogenicity in murine models [24]. The red-shifted bioluminescent strain, derived from the WT CEA10 strain, was kindly provided by Matthias Brock of the University of Nottingham, UK.

The ten additional *P. aeruginosa* strains used in this study are presented in Table 2. The strains were routinely cultured on brain–heart infusion agar (BHIA) plates composed of 37 g/L brain–heart infusion broth (BHI, Carl Roth, Karlsruhe, Germany), and 15 g/L technical agar (Biolife Italiana, Milano, Italy). Plates were incubated at 37 °C overnight in a humidified incubator with 5% CO_2_. 

### 2.2. Collection of Aspergillus spp. Conidia

Conidia were collected from two 10-day-old MEA plates. Briefly, 5 mL of filter-sterilized Dulbecco’s phosphate-buffered saline (DPBS, PAN-Biotech GmbH, Aidenbach, Germany), supplemented with 0.16% *v*/*v* Tween^®^ 80 (Sigma-Aldrich/Merck, Darmstadt, Germany) were added to each plate. The mycelial surface was rubbed using a delta cell spreader to dislodge and suspend conidia. The buffer containing conidia was collected in a 50 mL Falcon tube, and the entire procedure was repeated once more. Conidia suspensions were then filtered through a 40 µm cell strainer into a new 50 mL Falcon tube, and centrifugated at 2000× *g* for 10 min. Supernatant was removed and pellet was rinsed three times by resuspending it in 5 mL DPBS. Conidia were then counted with an improved Neubauer counting chamber and stored at 4 °C until use, for a maximum of 4 weeks. This method allows for the recovery of 300,000 to 800,000 conidia per microliter depending on the sporulation of the fungus in the culture plates used. Colony-forming unit (CFU) counting was also performed in order to correct for viable conidia.

### 2.3. Collection and Processing of Bronchoalveolar Lavage Fluid (BALF) Samples

The BALF samples used in this study were provided by the Lausanne University Hospital (CHUV) Division of Pulmonology and originated from a large longitudinal study in lung transplant recipients, which aimed to characterize the composition of the lung microbiota through metagenomic sequencing [27]. Subjects underwent transoral bronchoscopy as described in Das, Bernasconi [27]. Briefly, Bronchoalveolar lavage fluid (BALF) was collected via the following procedure: the bronchoscope was wedged either in the middle lobe or lingula of the allograft and 100–150 mL of normal saline was instilled in 50 mL aliquots that were pooled. BALF recovery was measured, and a fraction of 3 mL was stored at 4 °C and centrifuged within 3 h at 14,000× *g* for 10 min. The pellet was snap frozen and stored at −80 °C until further processing. A negative control obtained upon washing a ready-to-use endoscope with sterile saline was prepared following the same procedure. Glycerol samples of BALF were obtained for this study and were stored at −80 °C until use.

### 2.4. Enrichment and Isolation of Antagonistic Bacterial Strains from BALF Samples

Antagonistic bacterial strains inhibiting the germination and growth of *A. fumigatus* were enriched and isolated from BALF samples using a soft agar overlay plate assay with RPMI 1640 medium. RPMI medium was chosen because of the similarity in composition with the fluid lining the lung epithelium surface [27]. Briefly, 500 mL of filtered and sterilized, 2X RPMI 1640 liquid medium (10.4 g in 500 mL, Capricorn Scientific GmbH, Ebsdorfergrund, Germany, Cat. N° RPMI-A-P10) was mixed with 500 mL of 3% technical agar solution to make 1X RPMI 1640 1.5% agar medium which was then poured in 6 cm Petri plates. A second soft-agar (0.6%) layer of 2 mL of the same medium, in which 10 µL of *A. fumigatus* CEA10 (112,500 conidia/µL) were added, was poured on top of the 1.5% agar layer. While waiting for the soft agar layer to harden, 5 BALF samples (Table 3) were first thawed and pooled in a 15 mL Flacon tube and centrifuged at 3000× *g* for 10 min to concentrate bacterial biomass. Supernatant was removed until approx. 1 mL was left. Then, 250 µL of this BALF concentrate were spread onto the soft agar layer before the plates were incubated for 24 h in a humidified CO_2_ incubator. After incubation, individual strains were obtained through serial colony purification steps. 

### 2.5. Confrontation Assay on a Soft Agar Overlay Assay Plate

To qualitatively assess the inhibitory potential of each individual strain obtained, a similar bi-layered soft agar assay was used. A first layer of 1.5% agar RPMI medium was poured into 9 cm Petri dishes. Then, a second layer of 0.6% soft agar, containing 2 × 10^6^ *A. fumigatus* CEA10 conidia per 5 mL of medium, was poured on top of the first layer. Bacterial liquid cultures in BHI were set to an OD = 0.8, and five µL drops of these cultures were then spotted on top of the soft agar layer. Four different bacterial strains were spotted in a cross pattern on a single plate to maximize distance between each strain. Plates were incubated at 37 °C for 24 h.

### 2.6. Identification of the Bacterial Isolates

DNA samples were obtained by thermolysis. Briefly, bacterial biomass was sampled using a wooden toothpick and resuspended in 20 µL in PCR-grade water in a PCR tube. Then, PCR tubes were placed in a thermocycler for 10 min at 98 °C. The 16S rRNA gene was amplified using the ALLin™ HS Red Taq Mastermix, 2X kit (highQu, Kraichtal, Germany) with the GM3f (5′-AGAGTTTGATCMTGGC-3′)-GM4r (5′-TACCTTGTTACGACTT-3′) primer pair [28]. PCR reactions were performed in a final volume of 25 µL containing the following: 12.5 µL ALLin™ HS Red Taq Mastermix, 2X, 0.5 µL of each primer, and 1 µL of DNA sample. PCR reactions were run on an Arktik Thermo Cycler (Thermo Fisher Scientific, Waltham, MA, USA) with the following program: initial denaturation step at 95 °C for 1 min; followed by 40 cycles composed of a denaturation at 95 °C for 15 sec, annealing at 56 °C for 15 sec, and elongation at 72 °C for 30 s; and a final elongation step at 72 °C for 2 min. PCR products were verified by agarose gel electrophoresis and purified using a MultiScreen^®^ plate (Millipore Corporation, Burlington, MA, USA). DNA was quantified with a BR Qubit assay (Thermo Fisher Scientific, USA). Purified PCR amplicons were then sent to Fasteris, Life Science Genesupport SA (Plan-les-Ouates, Geneva, Switzerland) for Sanger sequencing. Nucleotide sequences were analyzed using NCBI nucleotide BLAST [29].

### 2.7. Genotypic Profiling of the Isolated P. aeruginosa Strains by Random Amplified Polymorphic DNA PCR (RAPD–PCR)

DNA samples were obtained by thermolysis as explained in Section 2.7. RAPD–PCR was performed to do a genotypic profiling of the five isolated strains of *P. aeruginosa* using primer 272 (5′-AGCGGGCCAA-3′). This primer was used due to its higher discriminatory potential for closely related genotypes compared with other primers used for *P. aeruginosa* genotyping [30,31]. RAPD–PCR reactions were performed in a final volume of 25 µL containing the following: 12.5 µL ALLin™ HS Red Taq Mastermix 2X (highQu, Germany), 4 µL of primer 272, and 1 µL of DNA sample. Reactions were run on an Arktik Thermo Cycler (Thermo Fisher Scientific, USA) with the following program: initial denaturation step at 95 °C for 2 min; followed by 35 cycles composed of a denaturation at 95 °C for 30 s, annealing at 35 °C for 30 s, and elongation at 72 °C for 2 min; and a final elongation step at 72 °C for 10 min. RAPD–PCR products were run on a 0.7% agarose gel at 70 V for 1 h.

### 2.8. High-Throughput Germination Screening Assay

A high-throughput (HTP) germination screening assay was developed to rapidly test bacterial isolates for their ability to inhibit the germination of conidia and the growth of *A. fumigatus*. To do so, the bioluminescent *A. fumigatus* strain CEA10b was incubated alone or in the presence of the five antagonistic bacterial isolates in a white-walled 96-well plate (Costar^®^ Corning, Corning, NY, USA, Ref. 3610). Each well contained a final volume of 200 µL of RPMI. For the co-culture, 80,000 *A. fumigatus* CEA10b conidia and 10 µL of bacterial liquid culture in RPMI (OD = 0.8) were added per well. The controls contained the same concentration of conidia or bacteria alone. An amount of 10 µL of 20 mM D-Luciferin solution (1 mM final concentration) was added in the well inoculated with conidia [32]. As positive control of germination inhibition, 2 µL of 50 mg/mL cycloheximide solution (in EtOH 99%) or 12.5 µg/mL voriconazole solution (in sterile MilliQ^®^ water, Merck KGaA, Darmstadt, Germany) were used. Optical density (at 600 nm) and bioluminescence (exposure time of 10 s) were measured every 30 min for 48 h following the incubation of the plates at 37 °C. Measurements were recorded using a Biotek Cytation 5 Cell Imaging Multimode microplate reader (Agilent, Santa Clara, CA, USA). All experiments were run in four independent replicates.

### 2.9. Confrontation on Calu-3 Cell Cultures on Permeable Transwell^®^ Inserts

Calu-3 cells were obtained from the American Type Culture Collection (Catalog No. ATCC HTB-55, Lot No. 70037771) and used from passages 17 to 21. Calu-3 cells were expanded in T-75 cell culture flasks with vent cap in MEM Eagle (PAN-Biotech GmbH, Aidenbach, Germany, Cat. No. P04-08056) supplemented with 10% of Panexin basic (PAN-Biotech GmbH, Aidenbach, Germany, Cat. No. P04-96950, Lot No. 3430322) and 1% of 100× Penicillin–Streptomycin (PAN-Biotech GmbH, Aidenbach, Germany, Cat. No. P06-07050). The cell cultures were maintained at 37 °C in a humidified incubator with 5% CO_2_. The culture medium was changed every other day. Cells were passaged when they reached 70–80% confluency by trypsinization with 0.05% Trypsin/0.02% EDTA (Lifeline Cell Technology, Frederick, MD, USA), and collected after centrifugation at 125× *g* for 5 min. Viable cells were counted with a hemocytometer using Trypan Blue PromoKine (PromoCell GmbH, Heidelberg, Germany, Cat. No. PK-CA902-1209).

Cells were seeded onto porous membranes with a density of 108,000 cells/well in 200 μL of culture medium in the apical side of Transwell^®^ inserts in 24-well plates (Corning, USA, Cat. No 3470). Transwell^®^ inserts were first coated with a solution of collagen type I (Ibidi, Gräfelfing, Germany, Cat. No. 50203) at a final concentration of 30 μg/mL prior to the seeding of the cells in order to allow proper cell attachment onto the porous membrane, as described in [33]. Afterwards, 600 μL of culture medium was added to the basolateral side of Transwell^®^ inserts in the 24-well plates. Transwell^®^ inserts (Art. N° 3470, Corning Incorporated, Corning, NY, USA) were placed in a humidified incubator at 37 °C with 5% CO_2_ for 15 days until confluence and formation of a monolayer. Medium was changed every other day as described previously. The cultures were observed daily using an EVOS™ XL Core bright field inverted microscope (Thermo Fisher Scientific, USA).

After 15 days of incubation, Transwell^®^ inserts were infected with either 100 conidia and/or bacterial cells per inserts. Prior to infection with the fungus, the apical medium was changed and 5 μL of IVISbrite D-Luciferin RediJect (30 mg/mL D-Luciferin, PerkinElmer, Shelton, CT, USA, Cat. No. 770504) was added per insert for bioluminescence imaging. Fungus- and bacteria-alone controls were included. Each condition was investigated in quintuplicate for the single and for the co-infections, and in octuplicate for the cell-only controls, in two independent experiments. All Transwell^®^ inserts were incubated in a humidified incubator at 37 °C with 5% CO_2_ for 19 h.

### 2.10. Immunofluorescence Staining and Confocal Imaging

Calu-3 cell cultures were fixed with 100 μL 4% paraformaldehyde in DPBS for 15 min at RT. Cells were then rinsed 3 times with 200 μL Tris Buffer Solution 1× pH 7.5 (TBS), with 5–7 min waiting time between each rinse. Cells were permeabilized with 100 μL 0.2% Triton X-100 in TBS for 15 min at RT and rinsed 3 times with 200 μL TBS, with 5–7 min waiting time between each rinse. After that, cells were blocked with 100 μL blocking solution (0.1% NaN_3_, 3% goat serum, 0.5% casein, 0.025% Tween 20 in TBS 1× pH 7.5, kindly provided by the Live Cell Imaging Core Facility, Department of Biomedical Research, University of Bern) for 30 min at RT. ZO-1 mouse anti-human conjugated with Alexa Fluor 594 (Thermo Fisher Scientific, USA, Cat. No. 339194) was prepared in the blocking solution (1/100). The actin stain (ActinGreen™ 488 ReadyProbes™ Reagent, Invitrogen, Carlsbad, CA, USA) and the nuclei counterstain (NucBlue™ Live ReadyProbes™ Reagent, Invitrogen, Carlsbad, CA, USA) were added to the same solution (2 drops/mL and 1 drop/mL, respectively). Anti-*Aspergillus* rabbit polyclonal primary antibody (Abcam, Cambridge, UK, Cat. No. ab20419) was also added (1/200) in the same solution for the conditions in which *A. fumigatus* conidia were inoculated. Fixed cells were incubated with the 100 μL solution containing the stains and antibodies overnight at 4 °C. The next day, fixed cells were washed 3 times with TBS supplemented with 0.025% Tween 20, with 5–7 min waiting time between each rinse. Goat anti-rabbit IgG secondary antibody (1/500) conjugated with Alexa Fluor 594 (Thermo Fisher Scientific, USA, Cat. No. A-11012) directed against Anti-*Aspergillus* antibody were prepared in TBS + 0.025% Tween 20. Fixed cells were incubated with 100 μL buffer containing the secondary antibodies for 3 h at RT in the dark. Cells were then washed 3 times with TBS + 0.025% Tween 20 for 7 min. Membranes from the inserts were then carefully cut out with a sharp knife, and mounted on a glass slide using the EMS shield mount mounting medium (Electron Microscopy Sciences, Hatfield, PA, USA) and imaged with a ZEISS Axio Observer Z1 LSM 980 with Airyscan 2 confocal microscope (Carl Zeiss AG, Oberkochen, Germany), using the C-Apochromat 40×/1.20 W Korr objective and Alexa Fluor 594 (red, λ_ex_: 590 nm, λ_em_: 610 nm), Alexa Fluor 488 (green, λ_ex_: 493 nm, λ_em_: 517 nm), and Alexa Fluor 405 (blue, λ_ex_: 401 nm, λ_em_: 422 nm) channels.

### 2.11. Assessment of Epithelial Barrier Function: TEER Measurements and Lucifer Yellow Permeability Assay

Trans-epithelial electrical resistance (TEER) was measured before media change using an EVOM3 Epithelial Volt/Ohm Meter with STX4 “chopsticks” electrodes daily for 15 days, starting from day 2 post-seeding. Two measurements were taken per inserts. TEER values were expressed in Ohms (Ω) per insert surface area (0.33 cm^2^).

Permeability of the Calu-3 monolayer was assessed using the paracellular marker Lucifer Yellow (LY, Lucifer Yellow CH Lithium salt, Biotium, Fremont, CA, USA, Cat. No. 80015). A 100 μg/mL LY solution was prepared in Hanks’ Balanced Salt Solution (HBSS, PAN-Biotech GmbH, Aidenbach, Germany, Catalog No. P04-32505) and 200 μL of this LY solution was added to the apical compartment, whilst 600 μL of HBSS was added to the basal compartment. Plates were incubated for 2 h 30 m in a humidified incubator at 37 °C with 5% CO_2_, and 100 μL of the basal solution were added to a black flat-bottom 96-well microplate. LY fluorescence intensity was measured with λ_exc_ = 485 nm and λ_em_ = 535 using a Biotek Cytation 5 Cell Imaging Multimode microplate reader (Agilent, USA). 

### 2.12. Bioluminescence Imaging

To assess the growth of *A. fumigatus* CEA10b on the Calu-3 monolayers when inoculated alone, or co-inoculated with the *P. aeruginosa* strains, bioluminescence imaging was performed using the Amersham Imager 680 (GE Healthcare, Chicago, IL, USA) in the chemiluminescence mode, with an exposure time of 5 min and binning of 32×.

### 2.13. Cytotoxicity Assay

Cytotoxicity was assessed through the quantification of dead cell-associated proteases using the fluorescent kit CytoTox-Fluor™ Cytotoxicity Assay (Promega, Madison, WI, USA, Cat. No. G9260) per the manufacturer’s instructions.

### 2.14. pH Measurements and Quantification of Calcium

pH and calcium concentration were quantified in the culture medium with the iSTAT 1 Blood Analyzer (Abbott, Abbott Park, IL, USA) using the CG8 + cartridges (Abbott, USA, Cat. No. 10002376).

### 2.15. Statistical Analyses

All graphical representations and statistical analyses were performed with RStudio (Version 2022.12.0 + 353) and GraphPad Prism 9 (Version 9.5.1). 

## 3. Results

### 3.1. Enrichment and Isolation of Lung Bacteria Inhibiting A. fumigatus Conidia Germination and Growth in One-Step Soft agar Plate Assay

A one-step soft agar plate assay was used to directly enrich and isolate bacterial strains able to inhibit the germination and growth of *A. fumigatus* CEA10 conidia from BALF samples. After 24 h of incubation, germination and growth of *A. fumigatus* conidia was visible in the fungus-alone control plate (Figure 2a). When the concentrated BALF was added onto the soft agar layer containing fungal spores, fungal growth was inhibited, and the fungus grew only in the area where bacteria were not present (Figure 2b). Moreover, an inhibition zone was visible in the interaction zone between bacteria and the fungus (double-headed arrow, Figure 2c).

Five macroscopically different bacterial isolates were obtained after the isolation step. The inhibitory action of the five isolates was confirmed by placing them in confrontation with *A. fumigatus* using the soft agar assay in RPMI medium. All five bacterial strains showed a clear and strong inhibitory effect on *A. fumigatus* conidia germination and growth, which was made visible by the inhibition halos around the bacterial colonies (Figure 3a). Moreover, close-up images of the interaction zones between the bacteria and the fungus showed an inhibition zone where no fungal growth was observed (Figure 3c–g), compared with the zones in which fungal growth was visible (Figure 3b). 

The five BALF isolates were identified as *Pseudomonas aeruginosa* after Sanger sequencing of the 16S rRNA gene (Table 4). 

### 3.2. Genotyping of the Five BALF P. aeruginosa Isolates

The genotypic diversity of the five BALF *P. aeruginosa* isolates was then compared using a quick genotypic fingerprinting by RAPD–PCR to check if the isolated strains were clonal or not, given that the BAL samples were pooled to concentrate bacterial biomass. The RAPD–PCR of the five *P. aeruginosa* strains isolated in this study revealed highly similar banding patterns on an agarose gel (Figure 4). This suggests that the isolates were clonal or highly similar genetically. This became evident when the isolates were compared with ten additional *P. aeruginosa* strains, which exhibited clearly different genotypes, except for isolate Pa-1. Indeed, isolate Pa-1 showed a similar banding pattern to isolates b1 to b5, suggesting this expectoration strain is genetically similar to the BALF isolates.

### 3.3. High-Throughput Germination Screening Assay

A high-throughput germination screening assay was developed to rapidly assess the inhibition potential of BALF bacterial isolates against *A. fumigatus* and select the best candidates for subsequent testing on Calu-3 cell monolayers. Using bioluminescence as a redout, conidia germination and hyphal growth can be monitored through light emission measurements thanks to the constitutive expression of the red-shifted firefly luciferase, which is under the glyceraldehyde-3-phosphate dehydrogenase promoter (P*gpdA*) in *A. fumigatus* CEA10b. Figure 5 shows the kinetic of *A. fumigatus* conidia germination and growth, through the measurement of bioluminescence over 48 h, alone or in confrontation with the five BALF *P. aeruginosa* isolates. When inoculated alone, *A. fumigatus* shows an exponential increase in bioluminescence signal after 10 h incubation indicating growth (Appendix A). The positive inhibition control, i.e., cycloheximide, initially inhibited the germination of *A. fumigatus* conidia. However, after approximately 30 h incubation, an increase in bioluminescence signal was recorded, indicating germination and growth of the fungus (Appendix A). Regarding the confrontation with the five BALF *P. aeruginosa* isolates, strain b1 showed a partial inhibition effect against *A. fumigatus*, which conidia were able to swell (Appendix A). On the other hand, strain b2 was able to completely inhibit the germination and growth of *A. fumigatus* (Appendix A). This was shown by a very weak bioluminescence signal. Finally, strains b3 to b5 showed a partial inhibition of *A. fumigatus* growth but allowed swelling of some of the conidia. Based on these results, BALF *P. aeruginosa* strains b1 and b2 were selected for the Calu-3 in-vitro confrontations. 

In addition, to assess the specificity of the novel one-step enrichment and isolation method presented in this study, we also tested ten additional *P. aeruginosa* strains that were isolated from expectoration samples in a medical diagnostics laboratory (Table 2). These strains exhibited varying degrees of inhibition against *A. fumigatus*, similar to the BALF isolates (Appendix A). Based on these results, we then tested the isolates b1 and b2 against isolates Pa-1 and Pa-2, the former allowing germination, while the latter does not, as well as isolates PaM-1 and PaM-4, with the first inhibiting the germination, and the second allowing both germination and hyphal growth, in order to be able to compare them (Appendix A). The expectoration isolate Pa-1 inhibited the germination of *A. fumigatus* conidia, allowing less conidial swelling compared with isolate b1 (Appendix A). All six selected strains tested demonstrated a similar inhibition ability against *A. fumigatus*. 

### 3.4. In-Vitro Confrontation of A. fumigatus with BALF P. aeruginosa Isolates on Calu-3 Cell Monolayers

The effect of inoculation of 100 *A. fumigatus* CEA10b conidia alone or in co-culture with 100 *P. aeruginosa* b1 or b2 cells was assessed in Calu-3 cell culture monolayers on Transwell^®^ inserts. We first assessed fungal growth by imaging bioluminescence emission by *A. fumigatus*. The infection template of each of the three independent plates used is depicted on the top in Figure 6a–c, with the corresponding images taken with the bioluminescence imager on the bottom. Detection of light emission by *A. fumigatus* is visible by the presence of a black spot in the fungus-infected cell inserts (C + F, Figure 6). However, in the co-infected inserts, i.e., C + F + b1 and C + F + b2, no bioluminescence was detected (Figure 6), suggesting a control of *A. fumigatus* growth by the bacterial isolates b1 and b2.

We then performed confocal imaging to confirm inhibition of *A. fumigatus* conidia germination by both *P. aeruginosa* b1 and b2 strains, when co-cultured together on the Calu-3 cell monolayers. After 16 days of incubation, the Calu-3 cells in the ‘no-infection control’ displayed a polygonal shape with the actin filaments clearly visible at the edges of the cells and expressed tight-junctions (ZO-1 protein in red) (Figure 7a). After 19 h of infection, *A. fumigatus* conidia were able to germinate and grow, as indicated by the presence of hyphae (in red) on top of the monolayer (Figure 7b). Moreover, the orthogonal views of the Z-stacks images of the Calu-3 monolayers infected with *A. fumigatus* showed that the fungus forms a dense hyphal layer at the surface of the epithelial monolayer (Appendix A), and that some hyphae penetrate the Calu-3 monolayer (Appendix A). Infection of the Calu-3 cells with *A. fumigatus* did not seem to cause significant damage to the monolayer (Figure 7c). This is, however, not the case with both *P. aeruginosa* isolates b1 and b2, which caused significant damage to the Calu-3 monolayer, as shown by the presence of holes in the epithelial monolayer (Figure 7d,e). Moreover, tight junction proteins were found to be preferentially expressed at the edges of the holes, while actin was not present (Figure 7d,e). In the case of the co-infection conditions, we could confirm that *P. aeruginosa* strains b1 and b2 were able to control the germination and growth of *A. fumigatus* conidia, since no hyphae were visible on the Calu-3 monolayers (Figure 7f,h, respectively). Moreover, similar to the conditions in which both *P. aeruginosa* were alone, the tight junction protein ZO-1 was preferentially more localized at the edges of holes induced by the bacteria, while actin seemed to be absent (Figure 7g,i). 

We also assessed the epithelial barrier function of the Calu-3 monolayers through trans-epithelial electrical resistance (TEER) measurements and an LY permeability assay. Figure 8a depicts the evolution of TEER over 16 days. TEER showed a regular increase in all conditions until around 2500 Ω/cm^2^ at day 14 and stayed constant around this value until day 15, when the infection occurred. Finally, 19 h post-infection, TEER stayed at around 2500 Ω/cm^2^ for the cells-alone control and the *A. fumigatus*-alone treatment and dropped at 1000 Ω/cm^2^ and below in the co-infection treatments. 

The LY permeability assay showed similar results. Indeed, compared with the cell-alone control (C), which is at around 1.2% (Figure 8b), the measured LY permeability was significantly higher in the C + b1, C + b2, C + F + b1 and C + F + b2 conditions. Moreover, we also observed that, in the presence of bacteria b2, the permeability was higher than with bacteria b1 (Figure 8b).

Figure 9 shows the assessment of cytotoxicity upon infection with *A. fumigatus* alone, or co-inoculated with *P. aeruginosa* strains b1 or b2. Compared with the cells alone (C), *A. fumigatus* did not seem to induce a strong cytopathic effect (C + F). The same is true when *P. aeruginosa* b1 is added alone, or together with *A. fumigatus*, on the cells (C + b1 and C + F + b1). However, in the presence of *P. aeruginosa* b2 alone (C + b2), or with *A. fumigatus* (C + F + b2), the measured fluorescence intensity is significantly higher than in the control cells.

## 4. Discussion

In this study, we presented a novel one-step enrichment and isolation method based on a soft agar overlay plate assay in which bacteria are directly in contact with *Aspergillus* spp. conidia. This method considers the interkingdom interactions between bacteria and *Aspergillus* spp. that would occur in the lung environment in-vivo [22,23,35]. It also allows easy observation of fungal growth inhibition by candidate antagonistic bacteria during enrichment, indicated by an inhibition halo around the bacterial inoculum. Traditionally, the enrichment and isolation steps are completely independent from the confrontation. Indeed, antagonistic bacterial strains are first isolated, and then tested against the fungal pathogen of interest in a second step [36,37]. Here, this entire approach is performed in one single step, and is thus directly enriching for bacterial strains that are antagonistic to the fungal pathogen of interest. While standard bacterial–fungal confrontation experiments rely on active fungal mycelium, our approach targets conidia [38], which form the infective stage in the developmental cycle of *Aspergillus* spp. [39]. Once *Aspergillus* conidia reach the lungs, they interact with various bacterial strains, which may inhibit their germination and subsequent growth into hyphae. Colonization of the lung tissue by *Aspergillus* spp. is a known risk factor for the development of invasive pulmonary aspergillosis [4]. While conidia germination is known to be influenced by microenvironmental factors [8,9,10,11,12,13,14], the specific contribution of the lung microbiota in the development of *Aspergillus* spp. is not yet well understood. 

We used five BALF samples from subjects who underwent a lung transplantation and were followed up at the Lausanne University Hospital (CHUV), as sources for potential bacterial isolates inhibiting the germination of *Aspergillus* conidia. Enrichment and isolation on bi-layered soft agar RPMI medium containing *A. fumigatus* conidia yielded five bacterial isolates all identified as *P. aeruginosa*. *P. aeruginosa* is an opportunistic pathogen and is commonly isolated in lung transplanted patients [40,41]. Interestingly, genotypic profiling through RAPD–PCR revealed that these five isolates were clonal, or at least highly genetically similar, compared with the ten additional *P. aeruginosa* strains isolated from expectoration samples. RAPD–PCR genotyping was chosen as a fast genotypic profiling technique for the differentiation of the five strains isolated in this study. This profiling technique is commonly used for discriminating closely related *P. aeruginosa* isolates. Moreover, the primer 272 used in this study was shown to have a higher discriminatory potential compared with other primers used for *P. aeruginosa* genotyping [30,31]. The predominance of this specific clonal lineage of *P. aeruginosa* over other genotypes suggests that the enrichment method used here may have exerted strong selective pressure due to the presence of *Aspergillus* conidia. 

RPMI medium, which is commonly used in host–pathogen studies, supports the growth of a diverse range of bacterial species, including those present in the lung microbiota. Das, Bernasconi [27] showed that several lung isolates, such as for instance *Micrococcus luteus*, *Staphylococcus aureus*, *Staphylococcus epidermidis*, *Streptococcus vestibularis*, and *Streptococcus oralis*, were able to grow in RPMI. Moreover, Sanchez-Rosario and Johnson [42] and Leonidou, Ostyn [43] respectively showed that *Streptococcus pneumoniae* and *Rothia mucilaginosa* were also able to grow in RPMI. 

Given the broad range of bacterial species supported by RPMI, the observed predominance of *P. aeruginosa* in our samples is likely influenced by specific factors associated with lung transplant patients. Notably, three out of the five BALF samples used in this study were obtained from patients who had received antibacterial treatment, which could have influenced the predominance of *P. aeruginosa* isolates. Antibacterials, as well as antifungals, are often used as prophylactic or therapeutic treatments in lung transplanted patients [44,45]. Das, Bernasconi [27] showed that there is a negative relationship between the number of antibiotics administered to transplanted patients and the imbalance of the lung microbiota, with *Pseudomonas* spp. being more prevalent when increasing the number of administered antibiotics. While the opportunistic pathogenic nature of *P. aeruginosa* offers limited value for direct clinical translation for the strains obtained here, the enrichment and isolation approach described in this study can be applied to other BALF or upper respiratory tract samples to isolate commensal bacterial antagonists with greater potential for future clinical application.

We assessed and confirmed the inhibitory potential of the five BALF *P. aeruginosa* isolates by using, on the one hand, the same bi-layered soft agar RPMI plate assay employed during the enrichment, and, on the other, a high-throughput germination screening assay. The high-throughput germination screening assay permitted us to rapidly identify interesting bacterial candidates able to inhibit the germination of *A. fumigatus* conidia to test in subsequent Calu-3 cell monolayer models. The bioluminescent strain of *A. fumigatus* CEA10 used in this study was constructed by Resendiz-Sharpe, da Silva [26] by inserting a codon-optimized thermostable-red-shifted firefly luciferase. As the expression of the red-shifted firefly luciferase is under the control of the constitutively active glyceraldehyde-3-phosphate dehydrogenase promoter (P*gpdA*), light emission is observed upon germination of conidia and hyphal growth [26]. *P. aeruginosa* strains b1 and b2 were selected because of their differential inhibitory effect: strain b1 allowed swelling of conidia, while strain b2 completely inhibited conidia germination, which was easily visible in the high-throughput germination screening assay. Their inhibitory potential did not differ compared with the other *P. aeruginosa* isolates tested in this study, i.e., strains Pa-1 to Pa-5 and PaM-1 to PaM-5. Indeed, the interaction between *A. fumigatus* and *P. aeruginosa* is well described and has been reviewed extensively [23,35]. This interaction is common in CF patients and relies mainly on iron [46]. Among other mechanisms, *Aspergillus* and *Pseudomonas* mutually inhibit each other through the secretion of gliotoxin or siderophores (hydroxamate), and pyoverdine or dirhamnolipids by *A. fumigatus* and *P. aeruginosa*, respectively [23,35]. *P. aeruginosa* is also known to antagonize other major fungal pathogens such as *Candida* spp. [47] and *Cryptococcus* spp. [48].

Furthermore, *A. fumigatus* CEA10 was confronted with *P. aeruginosa* strains b1 and b2 on Calu-3 cell monolayers. Inhibition of conidia germination and growth was then assessed by bioluminescence emission, as well as confocal microscopy. Moreover, the impact of these *Aspergillus–Pseudomonas* confrontations on epithelial barrier function and cytotoxicity was also investigated. We confirmed the inhibitory effect of both *P. aeruginosa* strains, i.e., b1 and b2, on the germination of *A. fumigatus* conidia and its mycelial growth on Calu-3 cell monolayers. Indeed, no bioluminescent signal was recorded when *A. fumigatus* conidia were co-inoculated with *P. aeruginosa* strains b1 and b2, as compared with when *A. fumigatus* was inoculated alone. To our knowledge, this is the first time a bioluminescent strain of *A. fumigatus* has been used in a cell culture model. As light emission is proportional to fungal growth, bioluminescence measurements could be used as a semi-quantitative approach to measure *A. fumigatus* colonization on an in-vitro lung cell culture model. The inhibitory effect of *P. aeruginosa* strains b1 and b2 on *A. fumigatus* growth was also confirmed through confocal microscopy. While fungal hyphae were visible when *A. fumigatus* was inoculated alone on Calu-3 cell monolayers, no visible growth was observed in the co-cultures with *P. aeruginosa* b1 and b2.

Moreover, *P. aeruginosa* strains b1 and b2, either inoculated alone, or co-inoculated with *A. fumigatus*, were found to cause significant damage to the Calu-3 cell monolayers. Holes were visible in the Calu-3 monolayers, especially in the presence of *P. aeruginosa* b2. These observations were confirmed by the cytotoxicity assay, which showed a high fluorescent signal in the presence of *P. aeruginosa* b2 alone, or co-inoculated with *A. fumigatus*, suggesting the release of high numbers of dead-cell proteases. *P. aeruginosa* is known to have a strong cytotoxicity effect in lung epithelial cells, notably through the induction of various forms of cell death, leading to cell lysis [49]. However, the exact underlying mechanism for higher cytotoxicity in strain b2 is currently unknown. 

Epithelial barrier integrity was assessed through TEER measurements over 16 days, as well as LY permeability assay as an end-point measurement. TEER was found to increase regularly until reaching a value of around 2500 Ω/cm^2^ at day 15, when the infection was complete. Typical TEER values for healthy Calu-3 cells range between 1000 and 2000 Ω × cm^2^ in submerged (liquid–liquid interface) conditions [50,51]. TEER values in the cell-control only and *A. fumigatus*-alone infected Calu-3 monolayers stayed constant around 2500 Ω/cm^2^. However, a strong decrease in TEER was observed when both *P. aeruginosa* strains were added to Calu-3 cells alone, or co-inoculated with *A. fumigatus. P. aeruginosa* has a strong cytotoxicity effect [49] and is known to have a disruptive effect on TEER [52]. Permeability assay using the paracellular marker LY showed similar results. However, TEER was found to be more sensitive. TEER is a strong indicator of the integrity of the barrier, even before the ability to measure it using permeability markers [53]. As shown in the co-culture with bacterial strains b1 and b2, the decrease in TEER values, increase in LY permeability, and visible damage to the Calu-3 cell monolayer observed under microscopy, collectively indicate a significant disruption of epithelial barrier integrity caused by the bacteria.

Here, we have presented an innovative and promising one-step enrichment and isolation method with which to directly confront bacterial strains with *A. fumigatus* conidia, one that offers significant advantages over traditional methods of bacterial strain isolation. This one-step method not only accelerates the identification of bacterial strains antagonistic to fungal pathogens, but also better reflects interkingdom interactions occurring between bacterial strains from the lung microbiota and lung fungal pathogens such as *A. fumigatus*. However, our study has several limitations, as follows: (1) the clinical samples originated from a single hospital center, which may affect the broader applicability of the findings and (2) BALF samples had to be pooled because of low microbial biomass, potentially obscuring inter-individual variability.

The implications of the findings highlighted in this study and their translatability to clinical practice are significant. First, commensal bacterial strains isolated using the method and showing a significant germination and growth inhibition of *A. fumigatus* could be the basis of a microbial-based therapeutic product for the management of *Aspergillus* infections [54]. Second, these microorganisms can also be used to investigate the interactions with current therapies to better understand drug resistance development.

## 5. Conclusions

To conclude, we successfully isolated five genotypically similar *P. aeruginosa* strains capable of inhibiting the germination and growth of *A. fumigatus* conidia from BALF samples obtained from lung transplant recipients. These bacteria were able to inhibit the germination and growth of *A. fumigatus* conidia in different experimental set-ups. Although *P. aeruginosa* is a known opportunistic lung pathogen and antagonist of *A. fumigatus*, the results presented here serve as proof of concept, validating the experimental pipeline of one-step enrichment and isolation using the RPMI soft agar bi-layered plate assay described in this manuscript. This approach opens up new avenues for the targeted enrichment of antagonistic bacterial strains against specific fungal pathogens.

## Figures and Tables

**Figure 1 microorganisms-12-02025-f001:**
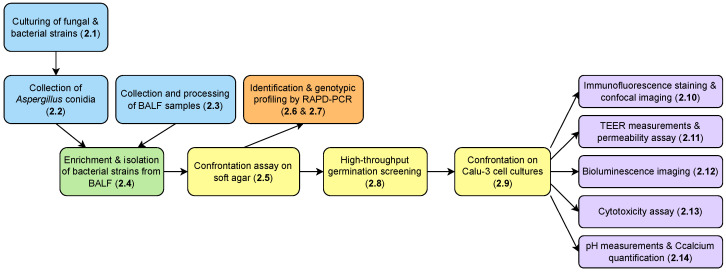
Flow chart diagram illustrating all of the methodological steps (Section 2.1, Section 2.2, Section 2.3, Section 2.4, Section 2.5, Section 2.6, Section 2.7, Section 2.8, Section 2.9, Section 2.10, Section 2.11, Section 2.12, Section 2.13 and Section 2.14) described in this study. Chart was produced using Draw.io (https://www.drawio.com/, accessed on 20 September 2024).

**Figure 2 microorganisms-12-02025-f002:**
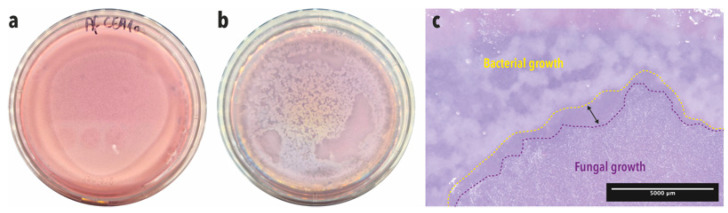
One-step soft agar enrichment and isolation of *A. fumigatus* conidia germination-inhibiting bacterial strains from BALF samples. (**a**) Fungus-alone control plate after 24 h incubation. (**b**) BALF-*Aspergillus* co-culture plate after 24 h incubation. (**c**) Close-up stereoscopic image of the co-culture plate where an inhibition zone is visible (double-headed arrow) in-between the bacterial growth zone, which is delineated with a yellow-dashed lined, and the fungal growth zone, which is delineated with a purple-dashed line.

**Figure 3 microorganisms-12-02025-f003:**
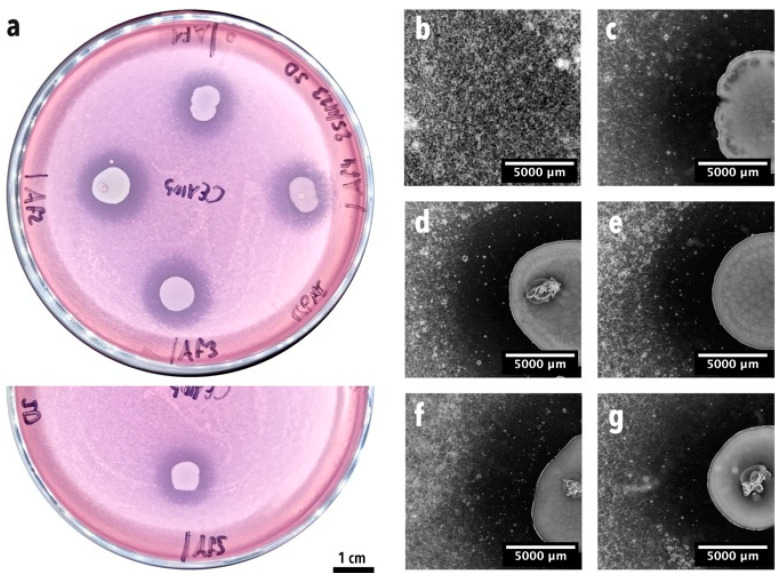
Soft agar confrontation assay between *A. fumigatus* CEA10 and the five BALF isolates. (**a**) Macroscopic image of the confrontation plates where *A. fumigatus* conidia germination inhibition halos are visible around bacterial colonies. Af1 = isolate b1, Af2 = isolate b2, Af3 = isolate b3, Af4 = isolate b4, Af5 = isolate b5. Images (**b**–**g**) were taken with a stereoscope. (**b**) Fungal growth control. (**c**) Isolate b1. (**d**) Isolate b2. (**e**) Isolate b3. (**f**) Isolate b4. (**g**) Isolate b5.

**Figure 4 microorganisms-12-02025-f004:**
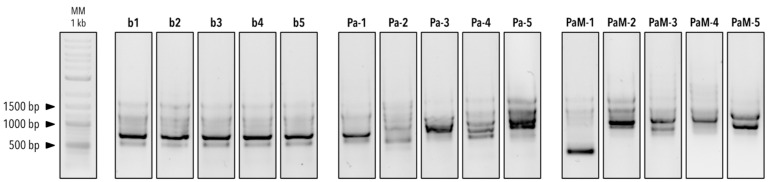
Genotypic profiling of the isolated *P. aeruginosa* strains by RAPD–PCR. MM: Molecular marker (Quick-Load^®^ Purple 1 kb Plus DNA Ladder (BioConcept Ltd., Allschwil, Switzerland)), b1–5 = *P. aeruginosa* strains isolated in this study, Pa-1–5 = *P. aeruginosa* strains, PaM-1–5 = cystic fibrosis *P. aeruginosa* strains.

**Figure 5 microorganisms-12-02025-f005:**
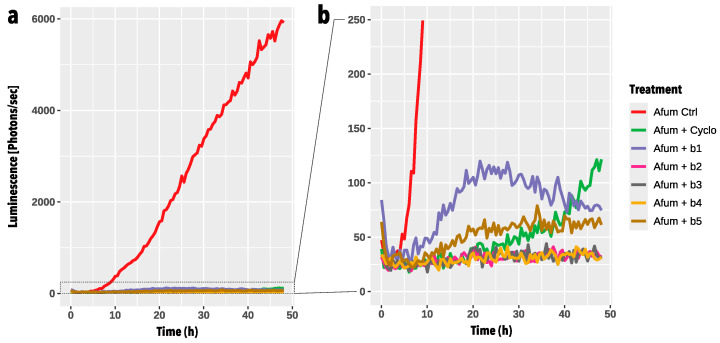
Evolution of *A. fumigatus* conidia germination and growth in the high-throughput germination screening assay in confrontation with the five *Pseudomonas aeruginosa* strains (b1–b5). The graph shows the mean bioluminescence signal [photons/sec] of four replicates emitted by *A. fumigatus* (Afum) over 48 h. Measurements were taken every 30 min. Panel (**b**) is a close-up view of (**a**). *A. fumigatus* alone (Afum Ctrl, in red) was used as a control for conidia germination and growth. *A. fumigatus* + Cycloheximide (Afum + Cyclo, in green) was used as a positive control of *A. fumigatus* inhibition. *A. fumigatus* (Afum) + b1–b5 represent the confrontation treatments with the five different isolates. Two-way ANOVA revealed that the treatment was highly significant (*p*-value < 2 × 10^−16^. Tukey multiple comparisons test was performed to compare means of Afum Ctrl with all other treatments (Afum + Cyclo, Afum + b1, Afum + b2, Afum + b3, Afum + b4, Afum + b5). All adjusted *p*-values (p adj) were highly significant (*p*-value = 0).

**Figure 6 microorganisms-12-02025-f006:**
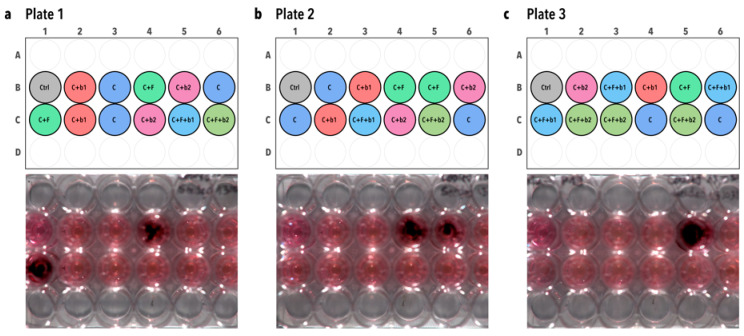
Bioluminescence imaging of infected plates. Plate layout is depicted on the top of each panel. Bioluminescence emission by *A. fumigatus* CEA10b alone or in the presence of the five BALF *P. aeruginosa* isolates was recorded and imaged using an Amersham Imager 680 (GE Healthcare, USA) in the chemiluminescence mode. Exposure time was set to 5 min and binning was set to 32×. (**a**) Plate 1. (**b**) Plate 2. (**c**) Plate 3. The black spots visible in the fungal infection condition, i.e., C + F, show bioluminescence emission by *A. fumigatus* CEA10b. Moreover, no bioluminescence emission was detected in the co-infections, i.e., C + F + b1 and C + F + b2, suggesting a biocontrol of *A. fumigatus* by the BALF isolates. The plate layout schemes were made in RStudio using the “ggplate” package [34]. C = cells alone. C + F = cells + *A. fumigatus*. C + b1 = cells + strain b1. C + b2 = cells + strain b2. C + F + b1 = cells + *A. fumigatus* + strain b1. C + F + b2 = cells + *A. fumigatus* + strain b2.

**Figure 7 microorganisms-12-02025-f007:**
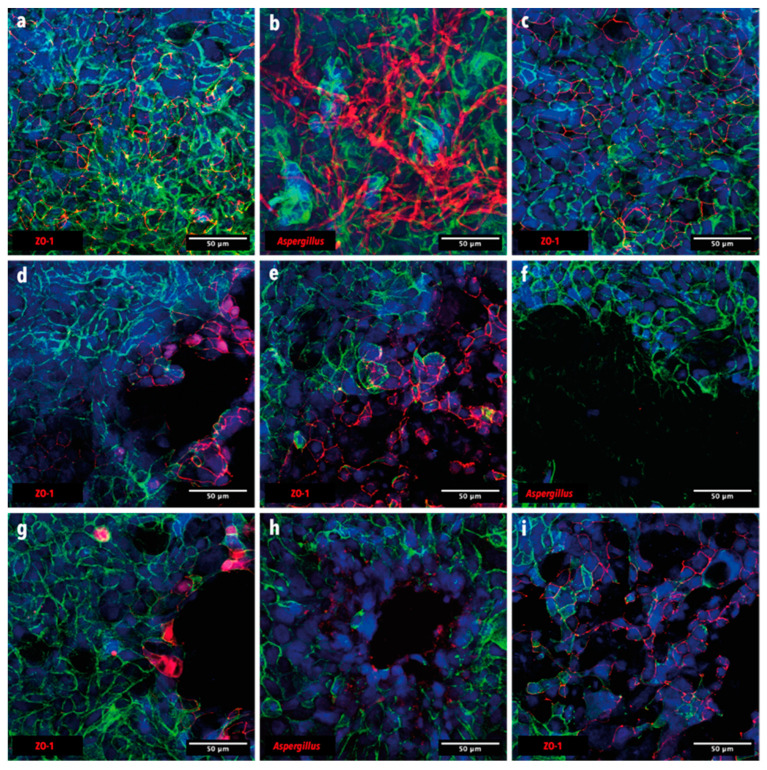
Confocal microscopic imaging. Actin is indicated in green, and nuclei in blue. (**a**) Cells alone control (ZO-1 antibody). (**b**) Cells + *A. fumigatus* (anti-*Aspergillus* antibody). (**c**) Cells + *A. fumigatus* (ZO-1 antibody). (**d**) Cells + b1 (ZO-1 antibody). (**e**) Cells + b2 (ZO-1 antibody). (**f**) Cells + *A. fumigatus* + b1 (anti-*Aspergillus* antibody). (**g**) Cells + *A. fumigatus* + b1 (ZO-1 antibody). (**h**) Cells + *A. fumigatus* + b2 (anti-*Aspergillus* antibody). (**i**) Cells + *A. fumigatus* + b2 (ZO-1 antibody).

**Figure 8 microorganisms-12-02025-f008:**
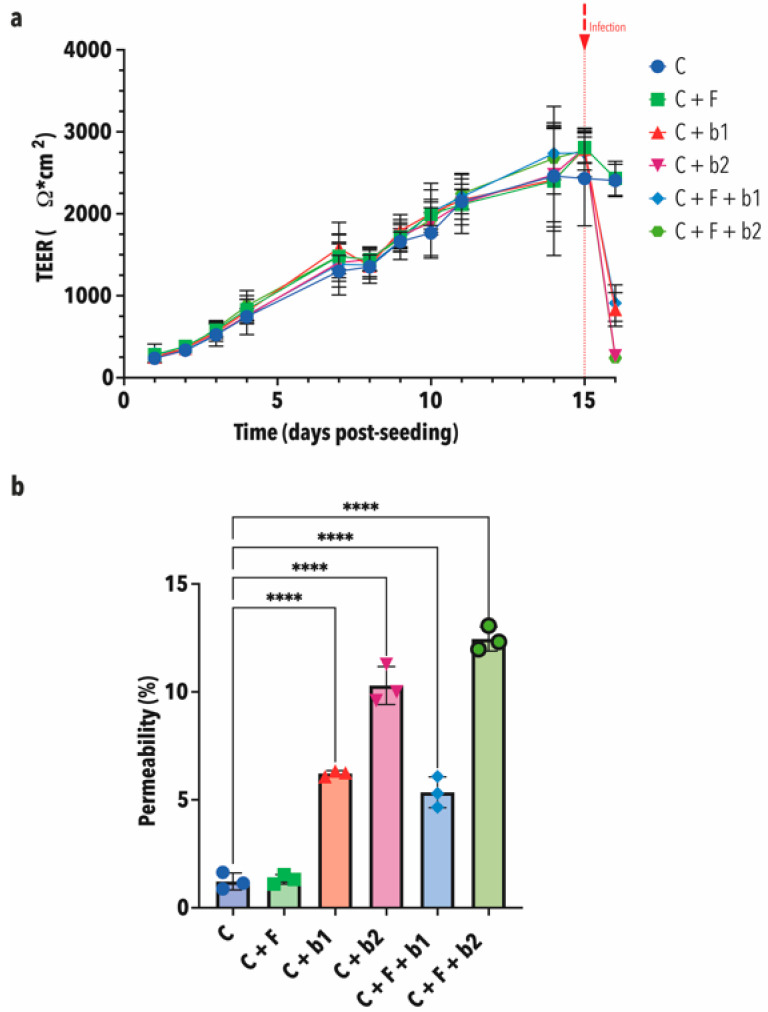
Epithelial barrier function. (**a**) TEER measurements. This graph shows the mean TEER in Ω × cm^2^ (+ - standard deviation) of five replicates. TEER was measured daily across Calu-3 monolayers for 15 days. TEER shows a constant augmentation until around 2500 Ω/cm^2^ at day 15. Infection was performed at day 16. TEER stayed at around 2500 Ω/cm^2^ for the cells-alone control and the *A. fumigatus*-alone treatment and dropped to 1000 Ω/cm^2^ and below in the co-infection treatments 19 h post-infection. (**b**) Lucifer Yellow (LY) permeability assay. Permeability is expressed as a percentage compared with the no-cell control, which is set to 100%. This graph shows the mean permeability in % (+ - standard deviation) of three replicates. Compared with the cell-alone control, which is at around 1.2%, the measured LY permeability was significantly higher in the C + b1, C + b2, C + F + b1 and C + F + b2 conditions. In the presence of bacteria b2, the permeability was higher than with bacteria b1. C = cells alone. C + F = cells + *A. fumigatus*. C + b1 = cells + strain b1. C + b2 = cells + strain b2. C + F + b1 = cells + *A. fumigatus* + strain b1. C + F + b2 = cells + *A. fumigatus* + strain b2. Statistical significance was tested by one-way ANOVA with multiple comparisons. Significance level used is the following: **** = *p* ≤ 0.0001.

**Figure 9 microorganisms-12-02025-f009:**
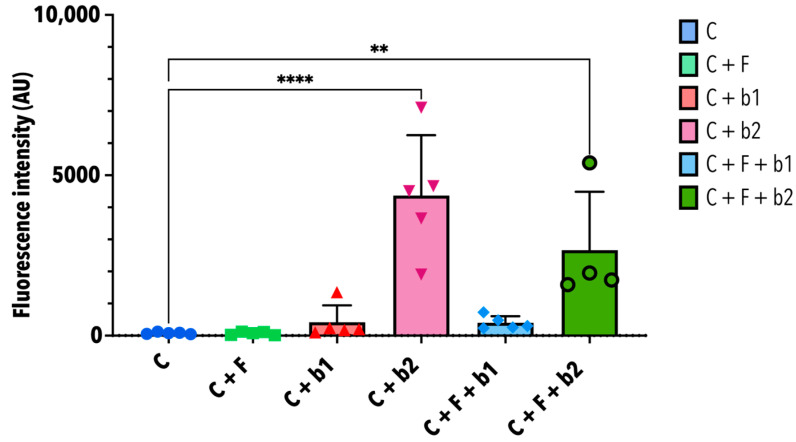
Cytotoxicity assay. This graph shows the mean fluorescence intensity in arbitrary units (AU) (+/- standard deviation) of five replicates Significant fluorescence intensity detected in the conditions with b2 compared with the control cells. C = cells alone. C + F = cells + *A. fumigatus*. C + b1 = cells + strain b1. C + b2 = cells + strain b2. C + F + b1 = cells + *A. fumigatus* + strain b1. C + F + b2 = cells + *A. fumigatus* + strain b2. Statistical significance was tested by one-way ANOVA with multiple comparisons. Significance levels used were the following: ** = *p* ≤ 0.01; **** = *p* ≤ 0.0001.

**Table 1 microorganisms-12-02025-t001:** *A. fumigatus* strains used in this study.

ID	Strain Name	Reference
CEA10	*Aspergillus fumigatus* CEA10, WT	[25]
CEA10b	*Aspergillus fumigatus* CEA10, red-shifted bioluminescent strain	[26]

**Table 2 microorganisms-12-02025-t002:** Additional *P. aeruginosa* strains used in this study. These strains were kindly provided by ADMED Microbiologie.

Strain Code	Species	Sample Origin
Pa-1	*Pseudomonas aeruginosa*	Expectoration
Pa-2	*Pseudomonas aeruginosa*	Expectoration
Pa-3	*Pseudomonas aeruginosa*	Expectoration
Pa-4	*Pseudomonas aeruginosa*	Expectoration
Pa-5	*Pseudomonas aeruginosa*	Expectoration
PaM-1	*Pseudomonas aeruginosa*, cystic fibrosis	Expectoration
PaM-2	*Pseudomonas aeruginosa*, cystic fibrosis	Expectoration
PaM-3	*Pseudomonas aeruginosa*, cystic fibrosis	Expectoration
PaM-4	*Pseudomonas aeruginosa*, cystic fibrosis	Expectoration
PaM-5	*Pseudomonas aeruginosa*, cystic fibrosis	Expectoration

**Table 3 microorganisms-12-02025-t003:** BALF samples used in this study. BALF samples from lung transplanted patients have been obtained from the Lausanne University Hospital (CHUV).

ID	Antibacterial Exposure	Antibacterial/Antifungal Drugs Administered
BAL1	Yes	Colifin (colistimethate)
BAL2	Yes	Bactrim (sulfamethoxazole + trimethoprim)
BAL3	Yes	Colifin (colistimethate)
BAL4	No	Ambisome (amphotericin B)
BAL5	No	Ambisome (amphotericin B)

**Table 4 microorganisms-12-02025-t004:** Identity of the isolated strains.

Strain Code	Species	Query Cover	Per. Ident
b1	*Pseudomonas aeruginosa*	100%	99.21%
b2	*Pseudomonas aeruginosa*	100%	97.40%
b3	*Pseudomonas aeruginosa*	100%	98.64%
b4	*Pseudomonas aeruginosa*	100%	98.72%
b5	*Pseudomonas aeruginosa*	100%	99.48%

## Data Availability

All data are available in the main text or the Appendix A. The R code used for the generation of kinetic bioluminescence graphic in the high-throughput assay can be found at https://github.com/palmierif/htp-biolum-afum (accessed on 30 September 2024).

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
