# Peer review of "One-Step Soft Agar Enrichment and Isolation of Human Lung Bacteria Inhibiting the Germination of Aspergillus fumigatus Conidia"

_microorganisms, 2024, doi:10.3390/microorganisms12102025_

Round 1

Reviewer 1 Report

Comments and Suggestions for Authors

It's a great manuscript. Congrats!

General comments:

Strengths: It's a novel and promising study. Good design and well presented. This novel one-step enrichment and isolation method offers advantages over other ones.

Weaknesses: Single center study.

Introduction. It is comprehensive, entertaining and informative. Includes 23 references, congrats!

Materials and methods, very complete and well explained. Maybe a flow chart diagram type of all the steps (2.1-2.14) could be interesting. Consider visiting this free-to-use tool:  https://www.yworks.com/products/yed

Discussion. At the end of this section, a paragraph is missing that refers to the strengths and weaknesses.

Conclusions. Perhaps it would be appropriate to include the conclusions in their own section.

References. 48 quotes are included, of which 30 are recent, that is, five years or less from their publication.

Reference 1. Is it complete? Idem quotes 3,7,9,16,23,33,37 and 46. Please, review all the references.

Reference 5. It would be possible to include a link.

Most citations indicate one author, other citations include two and some 3. Please, review.

Tables and figures. 4 tables, 8 figures and S6 figures. Everything is correct. Congrats!

Specific comments:

Discussion. First and second paragraph begin the same way. Please, review.

Author Response

It's a great manuscript. Congrats!

General comments:

Strengths: It's a novel and promising study. Good design and well presented. This novel one-step enrichment and isolation method offers advantages over other ones.

Weaknesses: Single center study.

Introduction. It is comprehensive, entertaining and informative. Includes 23 references, congrats!

Materials and methods, very complete and well explained. Maybe a flow chart diagram type of all the steps (2.1-2.14) could be interesting. Consider visiting this free-to-use tool:  https://www.yworks.com/products/yed

Thank you very much for this very nice suggestion. We prepared a flow chart diagram using Draw.io (https://www.drawio.com/) to illustrate all methodological steps described in our study. We included “All methodological steps outlined below (2.1. - 2.14.) are illustrated in the flow chart diagram depicted in Fig. 1” just after “2. Materials and Methods” in lines 110-111 followed by the flow chart below:

Figure 1. Flow chart diagram illustrating all the methodological steps (2.1. – 2.14.) described in this study. Done with Draw.io (https://www.drawio.com/).

All following figure numbers were changed accordingly.

Discussion. At the end of this section, a paragraph is missing that refers to the strengths and weaknesses.
Thank you very much for your comment. A section discussing the strengths and limitations of our study has been added to the end of the discussion in lines 698-707: “Here, we presented an innovative and promising one-step enrichment and isolation method which directly confronts bacterial strains with A. fumigatus conidia in a single step, offering significant advantages over traditional methods of bacterial isolation. This one-step method not only accelerates the identification of bacterial strains antagonistic to fungal pathogens, but also better reflects interkingdom interactions occurring between bacterial strains from the lung microbiota and lung fungal pathogens such as A. fumigatus. However, our study has several limitations: (1) clinical samples come from a single hospital center, which may affect the broader applicability of the findings, and (2) BALF samples had to be pooled because of low microbial biomass, potentially obscuring inter-individual variability.” 

Conclusions. Perhaps it would be appropriate to include the conclusions in their own section.
Thank you very much for this suggestion. We included the whole conclusive paragraph in section “5. Conclusion. To conclude, we successfully isolated five genotypically similar P. aeruginosa strains capable of inhibiting the germination and growth of A. fumigatus conidia from BALF samples obtained from lung transplant recipients. These bacteria were able to inhibit the germination and growth of A. fumigatus conidia in different experimental set-ups. Although P. aeruginosa is a known opportunistic lung pathogen and antagonist of A. fumigatus, the results presented here serve as proof-of-concept, validating the experimental pipeline of one-step enrichment and isolation using the RPMI soft agar bi-layered plate assay described in this manuscript. This approach opens up new avenues for targeted enrichment of antagonistic bacterial strains against specific fungal pathogens.”

References. 48 quotes are included, of which 30 are recent, that is, five years or less from their publication.

Reference 1. Is it complete? Idem quotes 3,7,9,16,23,33,37 and 46. Please, review all the references.

Thank you very much for this comment. All references have been reviewed.

Reference 5. It would be possible to include a link.

Thank you very much for this comment. A link has been added.

Most citations indicate one author, other citations include two and some 3. Please, review.

Thank you very much for this comment. All references have been reviewed to display only one author and “et al.” if more authors are listed.

Tables and figures. 4 tables, 8 figures and S6 figures. Everything is correct. Congrats!

Specific comments:

Discussion. First and second paragraph begin the same way. Please, review.

Thank you very much for pointing that out. “In this study” was removed at the beginning of the second paragraph.

Reviewer 2 Report

Comments and Suggestions for Authors

The manuscript " One-step soft agar enrichment and isolation of human lung bacteria inhibiting the germination of Aspergillus fumigatus conidia " aims to isolate and enrich bacterial strains from lung transplant recipients capable of inhibiting Aspergillus fumigatus conidia germination and growth using a novel enrichment method. Although the paper presents a certain scientific interest, there are some concerns regarding the data's validity and overall results. Here are some important comments:

1.     The manuscript mentions two fungi strains: A. fumigatus strains, these characteristics should be clarified in the following Table 1; In the context of clinical or environmental applications, experimentally validate this set of strains.

2.     The efficiency and reproducibility of dislodging these Aspergillus conidia using the delta cell spreader had to be clearly stated. Further validation of this technique may be required to compensate for inter- and intra-filter variability in spore count.

3.     The origin of bronchoalveolar lavage fluid (BALF) specimens seems to be somewhat restrictive. How consistent was the lung microbiome in patients? Describe the criteria for selectivity of BALF specimens and give more information about the turning point to match a good description that can be informative and representative.

4.     The confrontation assay refers to a 24-hour incubation, but I need quantitative data on inhibition zones or colony size, which is necessary for the qualitative assessment of this method. Statistical comparisons between bacterial strains (e.g. chi-squared analysis) would further strengthen our study.

5.     Explain why bioluminescence was chosen as a readout for germination inhibition. Would this signal also be influenced by things other than fungal germination (i.e., bacterial metabolism)? If direct microscopy or one of the second-line techniques is used, this will highly depend on the material used to confirm whether it has a schizonticidal effect.

6.     TEER measurements and Lucifer Yellow permeability assay measures However, contains more detail on typical TEER values expected in healthy and infected Calu-3 cells to help interpretation of epithelial barrier function.

7.     A detailed description of statistical methods used/not used isn't present in the manuscript (e.g., tests performed, sample size justification). Data: all data (especially those generated from confrontation assays, and high-throughput screenings) should be underpinned by sound statistical clarification to verify the significance.

8.     This is a critical part of the study; unfortunately, its description as a one-step assay in soft agar lacks specificity. Precise the agar concentration, bacterial inoculation density and incubation conditions to make it reproducible.

9.     Qualitative descriptors of the inhibition zones are used and further quantification information on the size of this could potentially strengthen results. It makes it possible to then do a statistical comparison of the given bacterial isolate.

10. Given that all isolates identified as P. aeruginosa, a more thorough discussion of the significance of these would be helpful. That all isolates may be the same species is surprising. Did they anticipate other bacterial species? Evaluating the impact on ecology.

11. The isolates were shown to be either clonal or very closely related genotypically by RAPD-PCR genotyping. Whole-genome sequencing or MLST would provide for more definitive identification, as RAPD-PCR may not have the discriminatory power required to separate strains of such close relatedness.

12. Data on damage to the Calu-3 cell monolayers of P. aeruginosa relevant strain(s). Some clarification is necessary as to whether the damage comes from indirect bacterial competition with A. fumigatus or a direct pathogenic effect of P. aeruginosa on epithelial cells. The authors note that such a distinction might be clinically relevant.

13. More detailed characterization of the bacterial isolates in inhibiting A. fumigatus germination may be warranted. Did they test these isolates against other fungi? They tried testing it on other clinically significant fungi like Candida spp might shed further light on the generality of these interactions.

14. Automated luminescent read-outs provide the same fungal growth but no statistical validation. Use statistical comparisons (e.g., ANOVA) between treated and control groups to confirm the inhibitory effects of distinct bacterial strains.

15. Cytotoxicity results in Strains b1 and e2 displayed the highest cytotoxic levels while strain a caused little cell death. Here, you should read up on why b2 has a higher cytotoxicity. Or is this due to some virulence factors or secreted products from b2?

16. Comment on the implications of this work for potential translation to a clinical setting, noting that P. aeruginosa is an opportunistic pathogen infecting patients with cystic fibrosis and other lung diseases. What might these results mean for future treatment or prevention?

17. Include a section discussing the limitations of the current study. For example, pooling the BALF samples might have obscured inter-individual variability. Suggest future experiments to address this issue, such as testing isolates from individual patients.

Comments on the Quality of English Language

The English language required minor editing.

Author Response

The manuscript " One-step soft agar enrichment and isolation of human lung bacteria inhibiting the germination of Aspergillus fumigatus conidia " aims to isolate and enrich bacterial strains from lung transplant recipients capable of inhibiting Aspergillus fumigatus conidia germination and growth using a novel enrichment method. Although the paper presents a certain scientific interest, there are some concerns regarding the data's validity and overall results. Here are some important comments:

  1. The manuscript mentions two fungi strains: A. fumigatus strains, these characteristics should be clarified in the following Table 1; In the context of clinical or environmental applications, experimentally validate this set of strains.

Both strains used in this study, i.e. wild-type A. fumigatus CEA10 strain, and its derived-red-shifted bioluminescent strain (CEA10b) are indicated in Table 1 with their respective reference. We indicated that the red-shifted bioluminescent strain was “derived from the WT CEA10 strain” in line 125. The wild-type A. fumigatus CEA10 strain is a clinically originated isolated widely used in laboratory testing (10.1093/mmy/myaa075).  The red-shifted bioluminescent strain has been kindly provided by Prof. Dr. Matthias Brock from the University of Nottingham (UK) who validated the corrected expression of the red-shifted luciferase. This information is now provided in the text in lines 123-124.

  1. The efficiency and reproducibility of dislodging these Aspergillus conidia using the delta cell spreader had to be clearly stated. Further validation of this technique may be required to compensate for inter- and intra-filter variability in spore count.

The protocol used for Aspergillus conidia suspension preparation is highly efficient and allows the recovery of 300’000 to 800’000 conidia per microliter depending on the sporulation of the fungus in the culture plates used. This information has been added in lines 151-152. Spore counting using a hemocytometer, as well as viable spore count by culture, was done for every new conidia suspension as indicated now in the same section.

  1. The origin of bronchoalveolar lavage fluid (BALF) specimens seems to be somewhat restrictive. How consistent was the lung microbiome in patients? Describe the criteria for selectivity of BALF specimens and give more information about the turning point to match a good description that can be informative and representative.

The BALF samples used in this study were provided by the Lausanne University Hospital (CHUV), Division of Pulmonology and come from a large longitudinal study in lung transplant recipients which aimed to characterize the composition of the lung microbiota through metagenomic sequencing (see 10.1038/s41467-021-22344-4). This has been added in lines 156-159. However, due ethical permit and data protection, no further information on pathologies can be provided.

  1. The confrontation assay refers to a 24-hour incubation, but I need quantitative data on inhibition zones or colony size, which is necessary for the qualitative assessment of this method. Statistical comparisons between bacterial strains (e.g. chi-squared analysis) would further strengthen our study.

The confrontation assay in the soft agar overlay plate (section 2.5) was used to assess qualitatively the inhibition of Aspergillus growth by the isolated bacterial strains. This has been clarified in the manuscript in line 191 to avoid misunderstandings. Quantitative assessment of the inhibitory potential was performed using the high-throughput germination screening assay described in 2.8.

  1. Explain why bioluminescence was chosen as a readout for germination inhibition. Would this signal also be influenced by things other than fungal germination (i.e., bacterial metabolism)? If direct microscopy or one of the second-line techniques is used, this will highly depend on the material used to confirm whether it has a schizonticidal effect.

Thank you for this comment. As explained in lines 642-643, “the expression of the red-shifted firefly luciferase is under the control of the constitutively active glyceraldehyde-3-phosphate dehydrogenase promoter (PgpdA)”. Therefore, bioluminescence can be monitored and measured overtime and is correlated with fungal growth and biomass, as “light emission is observed upon germination of conidia and hyphal growth”. This has been added in lines 422-426. The bioluminescent signal cannot be influenced by bacterial metabolism, since the luciferin substrate is only used by the luciferase-expressing A. fumigatus strain. Microscopic pictures for each confrontation conditions are available in supplementary figures S1, S3 and S5.

  1. TEER measurements and Lucifer Yellow permeability assay measures However, contains more detail on typical TEER values expected in healthy and infected Calu-3 cells to help interpretation of epithelial barrier function.

Thank you very much for this comment. Typical TEER values for healthy Calu-3 cells range between 1000 and 2000 W*cm2 in submerged (liquid-liquid interface) conditions. This is now added in lines 697-698. TEER values of infected Calu-3 cells will strongly depend on the conditions and on the specific pathogens used. As shown with bacterial strains b1 and b2, the decrease in TEER values, increase in Lucifer Yellow permeability, and visible damage to the Calu-3 cell monolayer observed under microscopy, collectively indicate a significant disruption of epithelial barrier integrity. This has been added in lines 705-709.

  1. A detailed description of statistical methods used/not used isn't present in the manuscript (e.g., tests performed, sample size justification). Data: all data (especially those generated from confrontation assays, and high-throughput screenings) should be underpinned by sound statistical clarification to verify the significance.

All statistical tests used were added to the legends’ figures together with statistical significance.

  1. This is a critical part of the study; unfortunately, its description as a one-step assay in soft agar lacks specificity. Precise the agar concentration, bacterial inoculation density and incubation conditions to make it reproducible.
    We apologize in advance, but we do not understand this comment as all the information about agar concentration, bacterial inoculum density and incubation conditions was already provided in section 2.5.
  2. Qualitative descriptors of the inhibition zones are used and further quantification information on the size of this could potentially strengthen results. It makes it possible to then do a statistical comparison of the given bacterial isolate.

As explained in our response to comment 4, the confrontation assay in the soft agar overlay plate (section 2.5) was used to assess qualitatively the inhibition of Aspergillus growth by the isolated bacterial strains. Quantitative assessment of the inhibitory potential was performed using the high-throughput germination screening assay described in 2.8.

  1. Given that all isolates identified as P. aeruginosa, a more thorough discussion of the significance of these would be helpful. That all isolates may be the same species is surprising. Did they anticipate other bacterial species? Evaluating the impact on ecology.

Thank you for this comment. As indicated in the discussion in lines 623-629, the RPMI medium used for the one-step enrichment and isolation of bacterial strains supports the growth of a diverse range of bacterial species from the lung microbiota, including non-P. aeruginosa commensal strains. Therefore, the predominance of P. aeruginosa isolates may be largely attributed to the antimicrobial treatments received by the patients, which likely reduced overall microbial diversity and favored the most competitive species. Moreover, as noted in lines 620-622, the isolation of a single genotype of P. aeruginosa suggests a strong selective pressure due to the presence of A. fumigatus conidia.

  1. The isolates were shown to be either clonal or very closely related genotypically by RAPD-PCR genotyping. Whole-genome sequencing or MLST would provide for more definitive identification, as RAPD-PCR may not have the discriminatory power required to separate strains of such close relatedness.

RAPD-PCR genotyping was chosen as a rapid genotypic profiling technique for the differentiation of closely related strains, and that is commonly used for the profiling of P. aeruginosa isolates. While we acknowledge the usefulness of whole-genome sequencing or MLST for more precise discriminatory power in discriminating closely-related species, the use of RAPD-PCR was sufficient for our study since the primer we used (primer 272, cf 2.7.) was shown to have a higher discriminatory potential in distinguishing closely related genotypes compared to other primers typically used for P. aeruginosa genotyping. This information has been added now in lines 610-620.

  1. Data on damage to the Calu-3 cell monolayers of P. aeruginosarelevant strain(s). Some clarification is necessary as to whether the damage comes from indirect bacterial competition with A. fumigatus or a direct pathogenic effect of P. aeruginosa on epithelial cells. The authors note that such a distinction might be clinically relevant.

Thank you for this comment. As indicated in lines 685-690, the damage observed in the Calu-3 cell monolayers in the conditions where P. aeruginosa strains b1 and b2 were either inoculated alone, or co-inoculated with A. fumigatus, comes from both direct and indirect pathogenic effect of P. aeruginosa.

  1. More detailed characterization of the bacterial isolates in inhibiting A. fumigatus germination may be warranted. Did they test these isolates against other fungi? They tried testing it on other clinically significant fungi like Candida spp might shed further light on the generality of these interactions.

The isolated bacterial strains using the soft agar overlay assay were only tested on A. fumigatus. However, P. aeruginosa is also known to antagonize Candida spp. (10.3390/jof5020034) and Cryptococcus spp. (10.3390/jof5020031). This has been added in lines 664-665.

  1. Automated luminescent read-outs provide the same fungal growth but no statistical validation. Use statistical comparisons (e.g., ANOVA) between treated and control groups to confirm the inhibitory effects of distinct bacterial strains.

Thank you for this comment. Statistical comparison has been performed and added to the legend of figures 5, S2 and S4.

  1. Cytotoxicity results in Strains b1 and e2 displayed the highest cytotoxic levels while strain a caused little cell death. Here, you should read up on why b2 has a higher cytotoxicity. Or is this due to some virulence factors or secreted products from b2?

Strain b2 displayed higher cytotoxicity, which may indeed be attributed to specific virulence factors or secreted products from this specific strain that enhance its pathogenic potential. However, we do not know the specific virulence factors or secreted products allowing for higher cytotoxicity than strain b1. This is now added in lines 692-693.

  1. Comment on the implications of this work for potential translation to a clinical setting, noting that P. aeruginosa is an opportunistic pathogen infecting patients with cystic fibrosis and other lung diseases. What might these results mean for future treatment or prevention?

Thank you very much for this comment. We completely agree that P. aeruginosa is a notable opportunistic lung pathogen, and therefore the results of this study on the inhibitory effect of P. aeruginosa strains on the germination and growth of A. fumigatus could not be directly translated to the clinical setting. However, the overall enrichment and isolation approach described in this study can be applied on other BAL or upper respiratory tract samples to isolate bacterial antagonists. This has been added in lines 639-643.

As mentioned in the discussion in lines 623-629, RPMI medium supports the growth of a diverse range of bacterial species from the lung microbiota, including non-P. aeruginosa commensal strains. These commensal bacterial strains, if showing a significant germination and growth inhibition of A. fumigatus, could then be at the basis of a microbial-based therapeutic product, i.e. live-biotherapeutic product, or their metabolic products, for the management of Aspergillus infections. Moreover, this microbial-based therapeutic product could be used as a combination therapy together with currently used antifungal compounds to enhance inhibition efficacy, and to reduce drug resistance development.

  1. Include a section discussing the limitations of the current study. For example, pooling the BALF samples might have obscured inter-individual variability. Suggest future experiments to address this issue, such as testing isolates from individual patients.

A section discussing the strengths and limitations of our study has been added to the end of the discussion in lines 710-719: “Here, we presented an innovative and promising one-step enrichment and isolation method to directly confront bacterial strains with A. fumigatus conidia that offers significant advantages over traditional methods of bacterial strains isolation. This one-step method not only accelerates the identification of bacterial strains antagonistic to fungal pathogens, but also better reflects interkingdom interactions occurring between bacterial strains from the lung microbiota and lung fungal pathogens such as A. fumigatus. However, our study has several limitations: (1) the clinical samples originated from a single hospital center, which may affect the broader applicability of the findings, and (2) BALF samples had to be pooled because of low microbial biomass, potentially obscuring inter-individual variability.”
